# The Impact of Insomnia on the Clinical Course and Treatment Outcomes of Rheumatoid Arthritis

**DOI:** 10.3390/biomedicines13102535

**Published:** 2025-10-17

**Authors:** Olivera Radmanović, Vladimir Janjić, Mirjana Veselinović, Aleksandar Kočović, Nemanja Murić, Milan Đorđić, Ermin Fetahović, Nikola Subotić, Anja Milojević, Milena Stojković, Elvis Mahmutović, Danijela Djoković, Branimir Radmanović

**Affiliations:** 1Clinic for Rheumatology and Allergology, University Clinical Center Kragujevac, 34000 Kragujevac, Serbia; olja.radmanovic@gmail.com (O.R.); miraveselinovic.m@gmail.com (M.V.); anjazanjac23@gmail.com (A.M.); 2Department of Internal Medicine, Faculty of Medical Sciences, University of Kragujevac, 34000 Kragujevac, Serbia; 3Department of Communication Skills, Ethics, and Psychology, Faculty of Medical Sciences, University of Kragujevac, 34000 Kragujevac, Serbia; vladadok@yahoo.com (V.J.); mcpikac@gmail.com (M.Đ.); erminfetahovic96@gmail.com (E.F.); 4Department of Psychiatry, Faculty of Medical Sciences, University of Kragujevac, 34000 Kragujevac, Serbia; milena-stojkovic94@live.com (M.S.); danijela-kckragujevac@hotmail.com (D.D.); biokg2005@yahoo.com (B.R.); 5Psychiatric Clinic, University Clinical Center Kragujevac, 34000 Kragujevac, Serbia; 6Department of Pharmacy, Faculty of Medical Sciences, University of Kragujevac, 34000 Kragujevac, Serbia; salekkg91@gmail.com; 7Department of Neuroscience, Faculty of Medical Sciences, University of Kragujevac, 34000 Kragujevac, Serbia; nikolasrf@gmail.com; 8Department of Biomedical Sciences, State University of Novi Pazar, 36300 Novi Pazar, Serbia; ehmahmutovic@np.ac.rs

**Keywords:** rheumatoid arthritis, insomnia, disease activity, inflammation, sleep quality

## Abstract

Background: Insomnia is markedly more prevalent in rheumatoid arthritis (RA) patients than in the general population and is closely linked to pain, fatigue, psychological comorbidities, and systemic inflammation. Evidence suggests a bidirectional relationship, where active disease worsens sleep quality, while poor sleep amplifies inflammatory activity and symptom severity. Methods: A narrative review was conducted using PubMed, Scopus, Web of Science, and Embase to identify studies from the last 15 years involving adult RA patients. Inclusion criteria required assessment of insomnia or sleep quality in relation to disease activity, treatment outcomes, or inflammatory markers. Data from clinical trials, cohort studies, and reviews were synthesized to examine prevalence, mechanisms, and therapeutic implications. Results: Insomnia affects up to 45% of RA patients and correlates with higher DAS28 scores, elevated CRP/ESR, increased pain sensitivity, and reduced quality of life. Contributing factors include chronic pain, stiffness, elevated IL-6 and TNF-α, depression, anxiety, and medication side effects. Conventional DMARDs, corticosteroids, and biologics indirectly improve sleep via inflammation control, with IL-6 inhibition showing potential sleep-specific benefits. Psychotropic agents may help in comorbid depression/anxiety but are best combined with cognitive behavioral therapy for insomnia (CBT-I). Conclusions: Insomnia is a prevalent, multifactorial problem in RA that adversely affects disease activity, symptom burden, and functional outcomes. Integrating sleep evaluation into routine RA management and adopting interdisciplinary strategies that address both inflammation and sleep disturbance may enhance patient outcomes. High-quality longitudinal studies using objective sleep measures are needed to clarify causal relationships and optimize therapy.

## 1. Introduction

Rheumatoid arthritis (RA) is a chronic, systemic autoimmune disease of unknown etiology, most commonly manifesting as chronic inflammatory arthritis. The condition primarily affects synovial joints, where inflammation of the synovial membrane initiates a cascade of pathological processes leading to pain, stiffness, swelling, and progressive joint destruction. Over time, this can result in significant functional impairment and disability of the affected limb. The pathogenesis of RA involves a complex interplay of genetic predisposition, environmental triggers, and immune system dysregulation, culminating in persistent inflammation and multisystem involvement [1].

The global prevalence of RA is generally estimated at 0.1% and 1% (most epidemiological studies assume approximately 0.5–1.0%), with a higher incidence in developed countries and among women and middle-aged adults [2,3]. Beyond musculoskeletal manifestations, RA often involves systemic complications, including cardiovascular, pulmonary, and neurological comorbidities, which further complicate disease progression and significantly reduce patients’ quality of life [4].

In recent years, nanomedicine has emerged as a promising strategy in the management of rheumatoid arthritis, aiming to improve drug delivery, reduce systemic toxicity, and enhance therapeutic efficacy. Examples include methotrexate-loaded nanoparticles and liposomal formulations, which allow for targeted delivery to inflamed joints and may improve bioavailability compared to conventional administration. Polymeric carriers and dendrimer-based systems have also been investigated for the sustained release of anti-inflammatory drugs. Despite these advances, nanomedicine-based approaches face important limitations, such as potential immunogenicity, high production costs, and challenges in large-scale clinical translation. Moreover, the majority of available studies remain preclinical or early-phase trials, and their long-term safety and comparative effectiveness are not yet well established. These aspects highlight both the potential and the current gaps in the application of nanotechnology in RA therapy [5,6].

Sleep disturbances, especially insomnia, are markedly more prevalent among individuals with RA compared to the general population. A systematic review and meta-analysis from 2024 found that the prevalence of insomnia in RA patients stands at around 45%, significantly exceeding that in the general population (~5.6%). Another meta-analysis highlighted that RA patients are four times more likely to experience insomnia than healthy individuals [7]. Mechanistically, factors such as chronic pain, nocturnal stiffness, elevated inflammatory cytokines, and psychological comorbidities (e.g., depression and anxiety) contribute to disrupted sleep patterns [8]. There is also evidence for a bidirectional relationship: disease activity exacerbates sleep problems, while poor sleep may amplify inflammatory responses and worsen RA symptoms [9].

Focusing on insomnia in RA is critical due to its dual impact on clinical symptoms and treatment outcomes. Insomnia is strongly linked to increased pain, fatigue, impaired daily functioning, and psychological burden. Furthermore, poor sleep may elevate pro-inflammatory cytokines such as IL-6 and TNF-α, central to RA pathogenesis. Conversely, effective disease control can improve sleep quality, highlighting the interdependence of disease activity and sleep. This complex interaction underscores the importance of assessing and managing insomnia to optimize therapeutic outcomes in RA. Addressing sleep disturbances as a therapeutic target supports an integrative approach, simultaneously controlling inflammation and improving sleep to enhance long-term disease management [10].

This narrative review systematically evaluates current evidence on the impact of sleep disturbances, particularly insomnia, on the clinical course and treatment outcomes of RA. It explores mechanisms linking sleep and disease activity, interactions between therapeutic interventions and sleep quality, and potential strategies to improve patient care. The review emphasizes the clinical relevance of routinely assessing and managing sleep in RA, which may enhance quality of life, reduce pain and fatigue, and improve overall treatment efficacy [11]. Additionally, this review identifies current gaps in the literature, particularly regarding causal relationships and the effects of novel therapies on sleep, providing direction for future research and the development of integrative approaches to RA management.

## 2. Methodology of Literature Review

This narrative review was conducted following systematic principles to ensure rigor and comprehensiveness. PubMed, Scopus, Web of Science, and Embase were searched for peer-reviewed articles, clinical trials, cohort studies, and review papers addressing sleep issues in RA patients. Only English-language publications from the past 15 years were included to focus on the most recent and relevant evidence.

The search strategy combined Medical Subject Headings (MeSH) and free-text terms using Boolean operators. Key search terms included: “rheumatoid arthritis,” “RA,” “sleep disorders,” “insomnia,” “sleep quality,” “disease activity,” “treatment outcomes,” and “fatigue.” The operators “AND” and “OR” were used to refine the search and identify relevant articles comprehensively.

Studies were included if they focused on adult patients diagnosed with RA according to established criteria (e.g., ACR/EULAR) and assessed sleep disturbances, insomnia, or overall sleep quality in the context of the disease. Additionally, studies needed to evaluate clinical outcomes, disease activity, or treatment response in relation to sleep. Excluded from the review were studies involving pediatric populations or other rheumatologic conditions, as well as abstracts, conference proceedings, non-peer-reviewed publications, or articles lacking sufficient data on sleep parameters or RA-related outcomes.

After removing duplicates, titles and abstracts were independently screened for relevance by two of the authors of this study. Full-text articles that met the inclusion criteria were then assessed for study rigor and for data extraction, using standardized criteria appropriate for each study design.

## 3. Sleep Disorders in Rheumatoid Arthritis

### 3.1. Prevalence and Types of Sleep Disorders in RA

Sleep disturbances are highly prevalent among patients with RA, with reported rates ranging from 40% to over 80%, depending on the population and assessment methods. Recent studies indicate that RA patients are four times more likely to develop insomnia compared to the general population, highlighting the need for increased attention to the diagnosis and management of these disorders [12,13].

Epidemiological data consistently demonstrate that insomnia affects approximately 40–45% of RA patients, with even higher rates in some cohorts. Risk factors include persistent joint pain, nocturnal stiffness, depression, anxiety, medication side effects, female sex, and longer disease duration. Identifying these risk factors is essential for timely intervention [13].

The most common sleep disturbances in RA patients include insomnia, fragmented sleep, restless legs syndrome (RLS), and obstructive sleep apnea (OSA). Insomnia often presents as difficulty initiating sleep, frequent nighttime awakenings, and non-restorative sleep, leading to increased daytime sleepiness and fatigue [13,14].

In addition to insomnia, RLS, also known as Willis-Ekbom’s disease, is prevalent among RA patients and is characterized by an uncomfortable sensation in the legs and an irresistible urge to move them, typically occurring in the evening. RLS frequently disrupts sleep and contributes to daytime sleepiness. Numerous studies have reported an elevated prevalence of RLS in RA, ranging from 27.7% to 31%, which is significantly higher than in the general population [15]. Notably, recent evaluations indicate that 90.8% of symptomatic RA patients were able to distinguish RLS-related leg discomfort from their arthritic symptoms, suggesting that these sensations are recognized as distinct from joint pain [15,16].

Obstructive sleep apnea (OSA), another common disorder in this population, is characterized by recurrent upper airway occlusion during sleep, leading to intermittent breathing pauses, reduced sleep quality, and excessive daytime sleepiness. While the general prevalence of OSA is estimated at 3–7% in men and 2–5% in women, several studies suggest that patients with RA are at increased risk. Population-based studies indicate a higher incidence of OSA in RA patients compared to non-RA controls, with some reports showing risks as high as 50% versus 31% in matched controls, highlighting the need for systematic assessment of sleep-disordered breathing in these patients [17].

It is important to note that the severity of sleep disturbances often correlates with disease activity, pain intensity, fatigue, and psychological comorbidities, underscoring the multidimensional impact of RA on sleep quality [3].

### 3.2. Mechanisms Underlying Insomnia in RA

The development of sleep disturbances in RA is multifactorial. Chronic pain and joint stiffness can disrupt sleep initiation and maintenance, while fatigue and reduced daytime activity contribute to circadian rhythm disturbances [3,18].

Psychological factors, such as depression and anxiety, further exacerbate sleep problems. Depression is the psychiatric condition most commonly associated with RA, with a meta-analysis reporting a prevalence of approximately 16.8% for major depressive disorder, though estimates vary widely due to differences in measurement methods and individual symptom courses [19]. Many RA patients also experience clinically significant depressive symptoms without meeting full diagnostic criteria, with prevalence estimates ranging from 14% to 48%. Clinically, comorbid depression in RA is linked to worsened pain, increased inflammation, greater disability, and overall poorer health outcomes, all of which can further disrupt sleep quality [20].

In addition, lifestyle factors, comorbid conditions, and the side effects of certain medications, including corticosteroids and some disease-modifying antirheumatic drugs (DMARDs), may also play a role in the pathogenesis of insomnia in RA patients [21].

### 3.3. Role of Inflammatory Cytokines in Sleep Regulation

Inflammatory processes inherent to RA are closely linked to sleep regulation. Pro-inflammatory cytokines, particularly interleukin-6 (IL-6) and tumor necrosis factor-alpha (TNF-α), are consistently elevated in RA and have been shown to disrupt normal sleep patterns. Poor sleep quality contributes to immune dysregulation in RA patients. Sleep deprivation elevates IL-6 and TNF-α, alters T helper cell balance, reduces natural killer cell activity, and disrupts hypothalamic–pituitary–adrenal axis function. These changes sustain synovial inflammation and may worsen disease activity, supporting a mechanistic model in which poor sleep and inflammation reinforce one another [22]. A meta-analysis including 14 studies with 890 RA patients and 441 healthy controls confirmed that serum IL-6 and TNF-α levels are significantly higher in RA patients, with this pattern observed across both Asian and Caucasian populations. Elevated cytokines can impair both slow-wave and rapid eye movement (REM) sleep, resulting in non-restorative sleep and increased daytime fatigue. This interplay suggests that systemic inflammation not only drives joint pathology but also directly contributes to sleep disturbances, establishing a self-perpetuating cycle that may further exacerbate disease activity and reduce overall quality of life [23].

Figure 1 illustrates the proposed mechanisms linking rheumatoid arthritis (RA) with insomnia, highlighting both biological and psychological pathways.

## 4. Impact of Insomnia on the Clinical Course of RA

### 4.1. Effect of Insomnia on Pain Perception and Stiffness

Insomnia amplifies the perception of pain and stiffness in patients with RA. Poor sleep can increase pain sensitivity and lower the pain threshold, further exacerbating subjective symptoms. Interestingly, in many RA patients, pain may exist out of proportion to peripheral inflammation, suggesting that central nervous system (CNS) pain amplification mechanisms, such as diminished conditioned pain modulation (CPM), play a role in enhancing pain perception [24,25].

Studies have shown that RA patients have lower CPM, pressure pain thresholds, and pain tolerance compared to healthy controls, not only at affected joints but also at non-joint sites. Sleep problems appear to partially mediate this relationship, indicating that insomnia may contribute to CNS sensitization and widespread hyperalgesia. These processes, combined with inflammatory mechanisms, can intensify joint pain and morning stiffness, leading to reduced functional mobility and a higher overall symptom burden [24,25].

### 4.2. Impact on Daily Functioning and Fatigue

Chronic insomnia contributes to increased fatigue and reduced daytime productivity in patients with RA. RA patients frequently report sleepiness, decreased concentration, and impaired ability to perform daily activities. Fatigue can manifest as an independent symptom, but when combined with pain and stiffness, it significantly diminishes quality of life [11].

A meta-analysis of 15 studies confirmed that RA patients have poor sleep quality, as measured by the Pittsburgh Sleep Quality Index (PSQI), with mean scores around 7, indicating clinically significant sleep disturbance. Poor sleep was strongly associated with lower quality of life (QoL), with each one-point increase in PSQI corresponding to a 2.4-point decrease in SF-36 QoL scores, affecting both physical and mental domains [26].

A cross-sectional study of 115 RA patients further demonstrated that sleep disturbances, depression, and higher disease activity were all significantly correlated with fatigue. Sleep problems alone explained a substantial portion of variability in fatigue, highlighting that insomnia is a key contributor to the daily functional limitations experienced by RA patients. These findings emphasize the importance of monitoring and managing sleep disturbances as part of comprehensive RA care, as improving sleep could potentially alleviate fatigue and enhance overall daily functioning [27].

### 4.3. Insomnia and Disease Activity (DAS28 and Other Indicators)

Insomnia in patients with RA is increasingly recognized as both a consequence and a potential driver of disease activity. Poor sleep has been associated with higher scores on established clinical measures, including DAS28 (Disease Activity Score of 28 joints), as well as elevated levels of inflammatory biomarkers such as C-reactive protein (CRP) and erythrocyte sedimentation rate (ESR). A study of 94 RA patients and 52 healthy controls evaluated sleep quality using PSQI and found that RA patients had significantly higher scores in subjective sleep quality, sleep latency, habitual sleep efficiency, sleep disturbance, and overall PSQI, reflecting marked sleep impairment. Analysis revealed that sleep disturbance was significantly associated not only with DAS28 and inflammatory markers (CRP, ESR), but also with pain, fatigue, depression, functional disability, radiological joint damage, quality of life, and duration of morning stiffness. These findings suggest that in RA patients, insomnia is not merely a secondary symptom but is intertwined with multiple aspects of disease burden. Addressing sleep disturbances may, therefore, improve quality of life, reduce fatigue and pain, and potentially enhance disease management by allowing a more accurate assessment of inflammatory activity and treatment efficacy [28,29].

### 4.4. Psychological Comorbidities: Depression, Anxiety, and Their Interaction with Sleep

Depression and anxiety are common comorbidities in patients with RA and frequently coexist with sleep disturbances. Insomnia can exacerbate psychological stress, while the presence of depression or anxiety can further impair sleep quality, creating a self-reinforcing cycle [30,31]. A prospective observational study conducted in Pakistan involving 169 RA patients and 85 healthy controls found that 71% of RA patients had psychiatric issues, compared with only 7.1% in controls. The mean depression score among RA patients was significantly higher (19.65 ± 1.44) than in controls (14.4 ± 1.31), and the mean anxiety score was also elevated (19.44 ± 2.4). Notably, 58.3% of RA patients were diagnosed with depression, 69.6% with major anxiety issues, and 27 patients had a mixed anxiety-depression disorder, predominantly depression-dominant. These findings illustrate the high prevalence and severity of psychological comorbidities in RA, highlighting their potential contribution to impaired sleep and overall disease burden [30].

A cross-sectional study of 209 RA patients in a tertiary rheumatology center in Romania further demonstrated that depression and anxiety are highly prevalent yet often underreported. While medical histories indicated prevalences of 10% and 8.1% for depression and anxiety, respectively, self-reported questionnaires revealed likely depression in 14.8–34.4% of patients and likely anxiety in 32.5%. Patients who screened positive for depression exhibited significantly higher DAS28 scores, increased tender and swollen joint counts, elevated patient global assessment, and more advanced functional stages. Similarly, patients with elevated anxiety scores were more frequently women and displayed higher tender joint counts and functional impairment. These findings emphasize that psychological comorbidities in RA are closely linked to higher disease activity, greater pain, and worse functional outcomes [31].

## 5. Therapeutic Aspects: Interaction Between Therapy and Insomnia

### 5.1. Conventional DMARDs and Their Impact on Sleep

Conventional disease-modifying antirheumatic drugs (DMARDs) are the foundation of RA therapy, as they slow disease progression, alleviate symptoms, and reduce the risk of joint damage and comorbidities. Conventional synthetic DMARDs (csDMARDs), including methotrexate, leflunomide, sulfasalazine, and antimalarials such as hydroxychloroquine, are typically used as first-line treatment, either as monotherapy or in combination, before escalation to biologic or targeted synthetic DMARDs if disease activity persists [32].

While csDMARDs are not specifically designed to target a single molecular pathway, their broad immunomodulatory effects can significantly improve sleep quality indirectly, primarily through reduction in inflammation, pain, and stiffness. Direct evidence linking csDMARDs to changes in sleep parameters is limited; improvements are generally attributed to better disease control rather than intrinsic sedative effects. However, adverse events such as gastrointestinal discomfort or neurological symptoms in some patients may negatively affect sleep. Careful selection of agents and monitoring for side effects remain important to optimize both disease management and sleep outcomes [33].

### 5.2. Corticosteroids and Effects on the Sleep–Wake Cycle

Over the decades, attitudes toward corticosteroid use in RA have shifted from initial enthusiasm to recognition of their significant adverse effects. Evidence from randomized trials and meta-analyses confirms that low to moderate doses (2.5–15 mg daily) can effectively reduce joint tenderness, pain, and swelling, and may even slow radiographic progression in early disease. However, these benefits are often short-lived, with rebound symptoms upon tapering, and the long-term impact on erosive damage remains inconsistent across studies [34,35].

Prolonged corticosteroid use, even at relatively low doses, carries risks including bone loss (most rapid in the first year), fractures, increased infection susceptibility, metabolic disturbances, cataracts, and muscle weakness. The difficulty of discontinuing therapy further complicates management, leading many rheumatologists to limit corticosteroids to the shortest effective duration [36]. Current guidelines generally recommend their cautious, intermittent use, prioritizing DMARD-based regimens for sustained disease control and minimizing cumulative steroid exposure [33,36].

Glucocorticoids, most notably prednisone, remain an important component of RA therapy due to their potent anti-inflammatory effects and rapid symptom control, particularly during acute flares or as bridging therapy until DMARDs take effect. Their impact on sleep is twofold: while symptom relief may indirectly improve sleep quality, glucocorticoids can disrupt circadian rhythms, delay sleep onset, and fragment sleep, especially when administered later in the day [34].

### 5.3. Psychotropic Agents Used in RA-Related Insomnia

#### 5.3.1. Overview of Psychotropic Medications

Psychotropic medications, including antidepressants, anxiolytics, and non-benzodiazepine hypnotics, are sometimes used to manage insomnia in RA, particularly when it coexists with depression or anxiety. Low-dose tricyclic antidepressants (e.g., amitriptyline) or sedating antidepressants (e.g., trazodone, mirtazapine) can improve both mood and sleep continuity, while selective serotonin reuptake inhibitors (SSRIs) may help treat underlying depression but can occasionally worsen sleep in sensitive individuals. Short-term use of non-benzodiazepine hypnotics (e.g., zolpidem, eszopiclone) may be considered for acute insomnia, but long-term use is generally discouraged due to tolerance and dependence risks [37]. In clinical practice, these agents are often combined with non-pharmacologic measures, most notably CBT-I, to achieve optimal and sustained improvements in sleep quality [38,39].

#### 5.3.2. Comparison with Existing RA Therapies

While antidepressants have been explored as adjunctive options for insomnia and mood disturbances in RA, their effects should be considered in relation to established RA therapies. Conventional synthetic DMARDs, such as methotrexate, sulfasalazine, and leflunomide, remain the cornerstone of RA treatment and primarily act by suppressing systemic inflammation. Biologic agents, particularly TNF inhibitors and IL-6 receptor blockers, offer targeted immunomodulation and have demonstrated robust efficacy in reducing disease activity and slowing radiographic progression. By comparison, antidepressants do not alter underlying inflammation but may improve sleep continuity, mood, and pain perception, thereby providing symptomatic relief. Their role is therefore best understood as complementary rather than disease-modifying [40].

#### 5.3.3. Considerations on Dosing Regimens

Another important limitation of the current literature is the lack of systematic evaluation of different dosing regimens. Optimization of dosage may influence not only the efficacy of antidepressants and psychotropic agents but also their tolerability in RA patients, who are often already receiving multiple concomitant medications. Similarly, fine-tuning the dose and timing of corticosteroids or DMARDs can have implications for sleep quality, particularly given the circadian influence of glucocorticoids on the sleep–wake cycle. Future clinical trials should systematically investigate dose–response relationships and circadian administration patterns, with the goal of maximizing efficacy while minimizing adverse effects [41].

#### 5.3.4. Economic Considerations

Finally, the economic dimension of RA management warrants attention. Biologic DMARDs, while highly effective, remain substantially more expensive than conventional DMARDs and psychotropic agents, posing challenges for healthcare systems and patient accessibility. Antidepressants, in contrast, are widely available at lower cost, but their use does not replace the need for effective inflammation control. Incorporating cost-effectiveness analyses into future research could help clarify the real-world viability of integrating sleep-focused therapies, such as antidepressants or cognitive behavioral interventions, into routine RA care [42,43].

#### 5.3.5. Gender Differences in Sleep Disturbances and Treatment Response

Most studies to date have not stratified insomnia outcomes in RA by sex, despite the fact that RA itself is more prevalent among women. Preliminary evidence suggests that female patients may experience higher rates of depression, anxiety, and sleep disruption compared to males. Future research should include sex-disaggregated analyses to clarify whether gender differences influence sleep quality, psychological comorbidities, or response to insomnia-targeted interventions in RA [44].

#### 5.3.6. Therapeutic Drug Monitoring (TDM) and Tapering Strategies

Therapeutic drug monitoring could be a valuable tool during tapering of insomnia treatments in RA, especially when using psychotropic medications. Regular monitoring of drug plasma levels and patient-reported outcomes may guide clinicians in minimizing withdrawal effects, preventing relapse of insomnia, and optimizing overall safety. Methodological challenges include inter-individual pharmacokinetic variability, polypharmacy, and the absence of standardized TDM protocols for most psychotropics [45,46].

#### 5.3.7. Biological and Psychological Mechanisms Linking Insomnia and RA

Insomnia may exacerbate RA activity through multiple overlapping pathways. On a biological level, sleep disruption elevates pro-inflammatory cytokines such as IL-6 and TNF-α, disrupts circadian rhythm regulation, and impairs immune homeostasis. On a psychological level, comorbid depression and anxiety may further impair sleep quality, creating a self-reinforcing cycle of poor rest and heightened disease activity [47,48].

### 5.4. Biological Treatments in RA: Implications for Sleep

Biological disease-modifying antirheumatic drugs (bDMARDs) have transformed the therapeutic landscape of rheumatoid arthritis (RA), offering targeted suppression of specific inflammatory mediators implicated in disease pathogenesis. Up to now, bDMARDs approved for RA therapy include agents with five different modes of action: TNF inhibition, T-cell co-stimulation blockade, IL-6 receptor inhibition, B-cell depletion, and interleukin-1 inhibition. These agents effectively reduce synovitis, slow radiographic progression, and improve functional outcomes [49].

In addition, targeted synthetic DMARDs (tsDMARDs), such as Janus kinase (JAK) inhibitors, and biosimilars are now also approved for RA, further expanding the options for personalized disease management. By directly modulating immune signaling pathways, these therapies not only control disease activity but may also indirectly influence factors affecting sleep, such as pain and systemic inflammation [50].

Clinical evidence consistently shows that biological therapies reduce disease activity, alleviate pain, and improve physical function. These improvements can indirectly support better sleep quality and daytime functioning in RA patients by reducing nocturnal discomfort and nighttime awakenings [51].

Although bDMARDs are not specifically designed to target sleep, it has been hypothesized that IL-6 inhibition may directly influence sleep architecture due to the cytokine’s role in inflammation and circadian regulation [51]. While biologic therapies may indirectly improve sleep through disease control, robust, controlled evidence confirming direct effects on sleep remains limited.

Most available studies assess sleep as a secondary or exploratory endpoint, often relying on self-reported questionnaires rather than objective measures such as polysomnography or actigraphy [52]. Furthermore, the heterogeneity in study populations, concomitant therapies, and outcome measures makes it difficult to draw firm conclusions about the direct impact of bDMARDs on sleep disturbances in RA [53].

Given the bidirectional relationship between sleep disturbances and inflammatory activity, future studies should systematically evaluate the effects of bDMARDs on sleep quality using standardized, validated tools. Longitudinal trials incorporating both subjective and objective sleep assessments could help clarify whether specific biological agents confer greater benefits for sleep outcomes. Additionally, exploring mechanisms, such as cytokine modulation, central pain processing, and circadian rhythm regulation, may uncover novel therapeutic strategies that address both inflammation and sleep disturbance in RA.

Table 1 summarizes the major RA therapies, their effects on disease activity, and their potential direct or indirect impact on sleep quality.

## 6. Evidence-Based Insights and Clinical Implications

Current evidence consistently indicates that sleep disturbances, particularly difficulty initiating or maintaining sleep, are considerably more frequent among individuals with RA compared to the general population [54]. The underlying mechanisms are multifactorial, encompassing persistent nocturnal pain and stiffness, elevated levels of pro-inflammatory cytokines, psychological distress, and, in some cases, medication-related adverse effects. A reciprocal relationship appears to exist between disease activity and sleep quality: more active disease tends to disrupt rest, whereas poor sleep may intensify systemic inflammation and aggravate RA manifestations. Beyond this, insomnia is closely linked to heightened pain perception, pervasive fatigue, and reduced functional capacity, thereby compounding the overall disease burden [55,56].

These findings highlight the importance of routinely assessing sleep quality as part of comprehensive RA management. Early identification and treatment of insomnia can potentially ease symptom severity, improve physical performance, and contribute to more stable disease control [57].

Effective management often requires an interdisciplinary approach, involving rheumatologists, mental health professionals, physiotherapists, and sleep medicine specialists. Alongside pharmacological interventions, non-drug strategies, such as CBT-I, structured physical activity, and patient education on sleep hygiene, should be integrated into care plans [38,58]. Certain biologic agents, through their potent anti-inflammatory effects, may indirectly improve nocturnal rest, though this remains an area requiring more targeted investigation [51].

However, much of the existing literature relies on cross-sectional designs and self-reported sleep assessments, which limit causal inference. Objective measurement tools, such as polysomnography or actigraphy, are seldom employed, and study cohorts are often small and demographically heterogeneous. In addition, important confounders, such as co-existing depression, anxiety, or restless legs syndrome, are not always adequately controlled for. The lack of standardized sleep assessment protocols further complicates comparisons across studies [59,60].

In clinical settings, it is advisable to implement systematic screening for sleep problems using validated instruments, with referral to sleep specialists when appropriate. Educational programs, individualized physical activity plans, and psychological support should be regarded as essential components of comprehensive RA care [58,61]. Future research should focus on large-scale, prospective investigations to clarify causal relationships, employ both subjective and objective sleep assessments for more accurate evaluation, explore the specific effects of various biologic and targeted synthetic DMARDs on sleep outcomes, and examine the efficacy of multidisciplinary treatment strategies that simultaneously address inflammation and sleep disturbances.

### Limitations and Future Directions

This review has several limitations that should be acknowledged. Most included studies were cross-sectional or observational in design, often with modest sample sizes and heterogeneous populations, which limits the ability to infer causal relationships between insomnia and RA outcomes. In addition, many studies relied solely on self-reported sleep measures rather than objective assessments such as polysomnography, introducing potential reporting bias. Important confounding factors, including depression, medication effects, and comorbid sleep disorders, were frequently insufficiently controlled. Furthermore, few studies stratified results by age, sex, or comorbidities, leaving key questions about subgroup differences unanswered.

Overall, the quality of the existing evidence is variable, with methodological limitations that constrain the strength of conclusions. Future research should focus on longitudinal, multicenter studies that integrate both subjective and objective sleep assessments and rigorously account for potential confounders. Randomized controlled trials evaluating pharmacological and non-pharmacological interventions for insomnia in RA are needed to assess their impact on disease activity, inflammatory markers, and quality of life. Personalized approaches, including consideration of gender-specific responses, optimized treatment dosing, and therapeutic drug monitoring, may enhance precision of care. Finally, incorporating health-economic analyses will help determine the cost-effectiveness and real-world feasibility of these interventions.

## 7. Conclusions

RA is often accompanied by sleep disturbances, particularly insomnia, which further exacerbate pain, fatigue, and functional limitations and may promote inflammatory processes. Recognizing and addressing sleep problems represents a crucial element of comprehensive patient care and has a significant impact on well-being and disease outcomes.

Although biologic therapies represent the newest and most costly form of RA treatment, data on their impact on sleep quality remain limited. Further research is needed to determine whether controlling inflammation with these agents can also lead to improvements in nocturnal rest and overall quality of life.

An integrative approach that combines effective inflammation control with targeted sleep support, involving collaboration among rheumatologists, sleep specialists, mental health professionals, and rehabilitation experts, offers the most promising framework for optimizing patient outcomes and overall health.

## Figures and Tables

**Figure 1 biomedicines-13-02535-f001:**
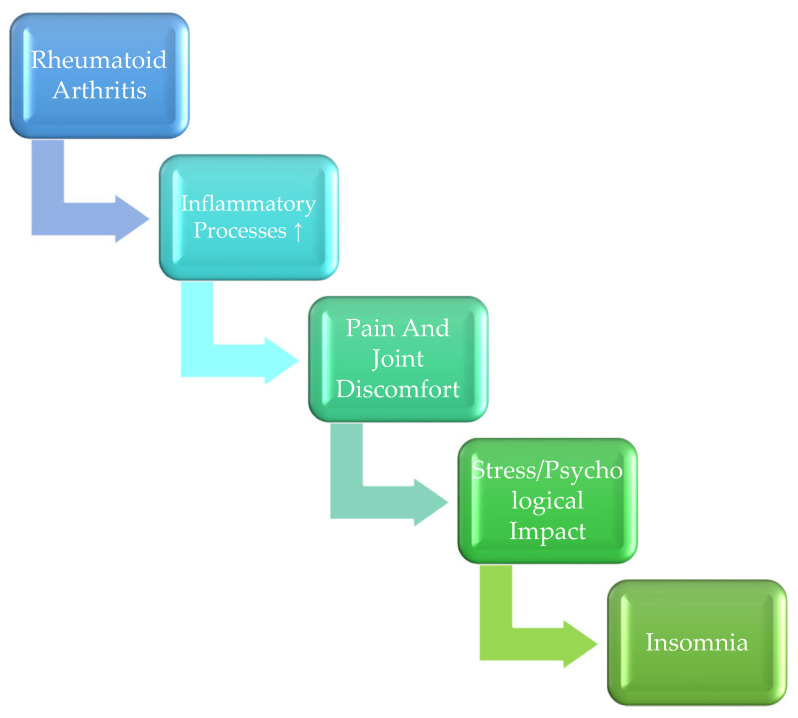
Proposed Mechanisms Linking Rheumatoid Arthritis (RA) and Insomnia.

**Table 1 biomedicines-13-02535-t001:** Overview of RA Therapies and Their Impact on Sleep.

Therapeutic Class	Examples of Drugs	Effect on Disease	Direct/Indirect Impact on Sleep	Limitations
csDMARDs	Methotrexate, Leflunomide, Sulfasalazine, Hydroxychloroquine	Reduction in inflammation, pain, and stiffness	Indirect (improved sleep through symptom control)	Gastrointestinal and neurological side effects may negatively affect sleep
Corticosteroids	Prednisone, Methylprednisolone	Rapid reduction in pain and inflammation	Directly may disrupt circadian rhythm and fragment sleep	Dose and timing are critical; long-term risks include osteoporosis and infections
bDMARDs	TNF inhibitors (Etanercept, Infliximab), IL-6 inhibitor (Tocilizumab), Rituximab	Targeted suppression of inflammation, slowing disease progression	Indirect improvement of sleep through inflammation and pain control	Limited direct evidence on sleep impact; cost and availability may be issues
tsDMARDs	JAK inhibitors (Tofacitinib, Baricitinib)	Targeted modulation of immune signaling	Indirect improvement of sleep	New drugs; long-term effects on sleep not well-studied
Psychotropic Agents	Trazodone, Mirtazapine, Amitriptyline, Zolpidem	Do not affect inflammation but improve sleep and mood	Direct improvement of sleep	Risk of tolerance and dependence

## Data Availability

Not applicable.

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
