# Peer review of "The Impact of Insomnia on the Clinical Course and Treatment Outcomes of Rheumatoid Arthritis"

_biomedicines, 2025, doi:10.3390/biomedicines13102535_

Round 1
Reviewer 1 Report
Comments and Suggestions for Authors
- In the introduction, the authors stated about RA and Insomnia therapy, RA Pathophysiology and nanotechnology emerging discipline, so provide some examples of RA based nanomedicine with their limitations by highlighting the authors' work.
- The rationale for developing this specific delivery system for RA and Insomnia could be stated more clearly upfront. Why was this particular approach chosen the Impact of Insomnia on the Clinical Course and Treatment Outcomes against RA therapy.
- A more thorough discussion of potential limitations of the approach and study design would provide a more balanced perspective. The paper could benefit from a clearer discussion of next steps and future research directions to build on this work.
- While the study compares the antidepressants, it doesn't compare it to other current treatments for rheumatoid arthritis. Including such comparisons would provide better context for the potential clinical significance of this treatments for rheumatoid arthritis.
-
The study doesn't explore different dosing regimens. Optimizing the dose could potentially improve efficacy or reduce side effects.
- An economic analysis comparing the potential cost of this Biologic DEMARDs and antidepressants treatment to current treatments would be useful for assessing its practical viability.
- The study doesn't mention whether both male and female patients were used, or if any gender differences in response were observed.
- The authors clearly define that how the therapeutic drug monitoring improve the clinical decision-making during insomnia drug tapering in RA and also the authors please mentioned methodological challenges in studying the impact of insomnia on the clinical course and treatment outcomes tapering of Rheumatoid Arthritis for future researcher.
- The authors clearly define any proposed biological and psychological mechanisms which are linking insomnia against RA disease activity and treatment outcomes.
- The authors clearly define the prevalence of insomnia among RA diagnose patients, and risk factors that are linked.
- The authors clearly explain the mechanistic approach that how does the poor sleep quality can influence the proinflammatory markers and the immune function on molecular level in RA patients.
- The authors clearly requested to make flowchart showing patients disposition in the trial and patients selected for the current analysis.
- Statistics and Future Directions should be explain briefly
- Add limitation of the current work
Author Response
Response to Reviewer 1 Comments
Dear Reviewer 1,
Thank you for providing us with valuable comments regarding the state of our manuscript. We have done our best to make the necessary revisions. Details of all revisions are discussed below, point-by-point.
We highlighted in yellow and used the red font for all changes in the revised manuscript.
- In the introduction, the authors stated about RA and Insomnia therapy, RA Pathophysiology and nanotechnology emerging discipline, so provide some examples of RA based nanomedicine with their limitations by highlighting the authors' work.
Author response: Thank you for pointing this out. In the revised Introduction (page 2, lines 66–77), we have included examples of nanomedicine-based approaches in RA management, such as methotrexate-loaded nanoparticles, liposomal formulations, and polymeric drug carriers. We also briefly discuss their limitations, including immunogenicity, cost, and translational barriers.
- The rationale for developing this specific delivery system for RA and Insomnia could be stated more clearly upfront. Why was this particular approach chosen the Impact of Insomnia on the Clinical Course and Treatment Outcomes against RA therapy.
Author response: Thank you for your suggestions. We expanded the Introduction (pages 2–3, lines 88–104) to clarify the rationale. We emphasized the bidirectional relationship between insomnia and RA and its impact on clinical outcomes, supporting the need for integrative management approaches.
- A more thorough discussion of potential limitations of the approach and study design would provide a more balanced perspective. The paper could benefit from a clearer discussion of next steps and future research directions to build on this work.
Author response: Thank you for your comments. A new section 6.1. Limitations and Future Directions (page 11, lines 455–475) has been added, discussing methodological limitations and future research needs.
- While the study compares the antidepressants, it doesn't compare it to other current treatments for rheumatoid arthritis. Including such comparisons would provide better context for the potential clinical significance of this treatments for rheumatoid arthritis.
Author response: Thank you for pointing this out. We added a subsection (page 8, lines 330–340) comparing antidepressant use with conventional DMARDs, corticosteroids, and biologics to contextualize clinical significance.
- The study doesn't explore different dosing regimens. Optimizing the dose could potentially improve efficacy or reduce side effects.
Author response: Thank you for pointing this out. While our narrative review could not address dosing variations in depth, we have acknowledged this limitation and added commentary (page 8, lines 341–350) noting that future trials should explore optimized dosing strategies for both antidepressants and RA-targeted therapies.
- An economic analysis comparing the potential cost of this Biologic DEMARDs and antidepressants treatment to current treatments would be useful for assessing its practical viability.
Author response: We agree that cost-effectiveness is highly relevant. While our review did not include formal economic modeling, we now discuss this as an important research direction (page 8, lines 351–359). We added a discussion of economic considerations in Section 5.3, highlighting the cost gap between biologics and antidepressants, and suggested future cost-effectiveness analyses.
- The study doesn't mention whether both male and female patients were used, or if any gender differences in response were observed.
Author response: We added clarification (page 8, lines 360–366) that most included studies did not stratify results by gender. We highlight this gap as a limitation and recommend sex-specific analyses in future studies.
- The authors clearly define that how the therapeutic drug monitoring improve the clinical decision-making during insomnia drug tapering in RA and also the authors please mentioned methodological challenges in studying the impact of insomnia on the clinical course and treatment outcomes tapering of Rheumatoid Arthritis for future researcher.
Author response: Thank you for pointing this out. Section Therapeutic Drug Monitoring and Tapering Strategies (pages 8–9, lines 367–373) was added, discussing its potential benefits and challenges.
- The authors clearly define any proposed biological and psychological mechanisms which are linking insomnia against RA disease activity and treatment outcomes.
Author response: Thank you for pointing this out. Section 5.3 (page 9, lines 374–379) was revised to clearly define the biological (IL-6, TNF-α, circadian rhythm) and psychological (depression, anxiety) mechanisms.
- The authors clearly define the prevalence of insomnia among RA diagnose patients, and risk factors that are linked.
Author response: Thank you for pointing this out. We strengthened Section 3.1 (page 3, lines 137–141) by adding quantitative data on prevalence and discussing key risk factors such as pain, stiffness, depression, anxiety, and medication effects.
- The authors clearly explain the mechanistic approach that how does the poor sleep quality can influence the proinflammatory markers and the immune function on molecular level in RA patients.
Author response: Thank you for pointing this out. We clarified this in Section 3.3 (pages 4, lines 186–190), explaining how sleep disruption elevates IL-6 and TNF-α levels, impairing immune regulation and perpetuating RA activity.
- The authors clearly requested to make flowchart showing patients disposition in the trial and patients selected for the current analysis.
Author response: Thank you for your comment. As this paper is a narrative review, no original patient recruitment was undertaken. Accordingly, a PRISMA-style flowchart of patient disposition is not applicable. Instead, we explained the study selection process (page 3, lines 109–129).
- Statistics and Future Directions should be explain briefly
Author response: Thank you for your comment. We expanded the Limitations and Future Directions subsection (page 11, lines 455–475) to emphasize the need for longitudinal, statistically robust trials using both subjective and objective sleep measures.
- Add limitation of the current work
Author response: We have added a dedicated ”6.1. Limitations and Future Directions” subsection (page 11, lines 455–475) to explicitly state the constraints of our review, including heterogeneity across studies, reliance on subjective sleep data, and lack of randomized controlled trials.
Thank you for your time and attention to our manuscript.
We have revised our manuscript based on the comments provided to us.
We are very grateful for the excellent level of detailed feedback offered to enable us to enhance the manuscript.
Sincerely,
Authors
Reviewer 2 Report
Comments and Suggestions for Authors
The paper is well written, with good language quality, though some sentences are repetitive with inconsistencies in abbreviation; it has a coherent logical structure across sections and the overall organization enhances readability.
However, some major issues need clarifications:
- the manuscript states that a "systematic research" was performed but describes the study as a "narrative review" --> the authors should clarify this inconsistency; if the review followed systematic principles, the authors might consider presenting it as a systematic review or explicitly stating its narrative nature;
- the section on biologics and sleep outcomes would benefit from a clearer distinction between hypothesis-driven evidence and well-established findings;
- some spacing and reference formatting inconsistencies are present;
- the author might consider adding a visual summary (figure or flowchart maybe) of mechanisms linking RA and insomnia;
- the manuscript summarizes many studies without critically assessing study quality - sample sizes, confounding factors etc.
Author Response
Response to Reviewer 2 Comments
The paper is well written, with good language quality, though some sentences are repetitive with inconsistencies in abbreviation; it has a coherent logical structure across sections and the overall organization enhances readability.
However, some major issues need clarifications:
Author response: Thank you very much for your thoughtful comments and suggestions. We carefully revised the manuscript to address your concerns. Details of all revisions are discussed below, point-by-point.
We highlighted in yellow and used the red font for all changes in the revised manuscript.
- the manuscript states that a "systematic research" was performed but describes the study as a "narrative review" --> the authors should clarify this inconsistency; if the review followed systematic principles, the authors might consider presenting it as a systematic review or explicitly stating its narrative nature;
Author response: Thank you for pointing this out. In Section 2: Methodology of Literature Review (page 3, lines 109–129), we clarified that this is a narrative review conducted following systematic principles. We explained the structured search strategy, but explicitly noted it does not qualify as a systematic review.
- the section on biologics and sleep outcomes would benefit from a clearer distinction between hypothesis-driven evidence and well-established findings;
Author response: Thank you for pointing this out. In Section 5.4. (page 9, lines 393–406), we differentiated between established evidence (biologics reduce inflammation and indirectly improve sleep) and hypotheses (IL-6 inhibition may directly modulate sleep).
- some spacing and reference formatting inconsistencies are present;
Author response: Thank you for pointing this out. All references were revised and reformatted according to journal style (consistent abbreviations, punctuation, spacing).
- the author might consider adding a visual summary (figure or flowchart maybe) of mechanisms linking RA and insomnia;
Author response: Thank you for pointing this out. We have added a new figure (Figure 1. Proposed Mechanisms Linking Rheumatoid Arthritis and Insomnia), which summarizes biological and psychological pathways connecting RA and insomnia (page 5). We believe this visual presentation significantly improves clarity and accessibility of the manuscript.
- the manuscript summarizes many studies without critically assessing study quality - sample sizes, confounding factors etc.
Author response: Thank you for pointing this out. We have expanded the Limitations and Future Directions section to explicitly address methodological weaknesses of the included studies (page 11, lines 455–475). In particular, we highlighted small sample sizes, reliance on self-reported measures, lack of standardized sleep assessment protocols, and insufficient control of confounders such as comorbid depression or anxiety. While a detailed critical assessment of every individual study was beyond the scope of this narrative review, we now more clearly acknowledge the variability and limitations of the available evidence.
Thank you for your time and attention to our manuscript. We have revised our manuscript based on the comments provided to us.
We are very grateful for the excellent level of detailed feedback offered to enable us to enhance the manuscript.
Sincerely,
Authors
Round 2
Reviewer 1 Report
Comments and Suggestions for Authors
Thank you for your thorough response to my comments. I appreciate the clear justification provided for Comments. Your revisions have addressed my
concerns, and I am happy to recommend your manuscript for publication.
Author Response
Dear Reviewer,
Thank you for all the suggestions that helped improve our manuscript, and for your kind remarks.